# Trap-and-Track for Characterizing Surfactants at Interfaces

**DOI:** 10.3390/molecules28062859

**Published:** 2023-03-22

**Authors:** Jeonghyeon Kim, Olivier J. F. Martin

**Affiliations:** Nanophotonics and Metrology Laboratory, Swiss Federal Institute of Technology Lausanne (EPFL), 1015 Lausanne, Switzerland

**Keywords:** optical tweezers, surfactants, adsorption, micelles, single-particle tracking

## Abstract

Understanding the behavior of surfactants at interfaces is crucial for many applications in materials science and chemistry. Optical tweezers combined with trajectory analysis can become a powerful tool for investigating surfactant characteristics. In this study, we perform trap-and-track analysis to compare the behavior of cetyltrimethylammonium bromide (CTAB) and cetyltrimethylammonium chloride (CTAC) at water–glass interfaces. We use optical tweezers to trap a gold nanoparticle and statistically analyze the particle’s movement in response to various surfactant concentrations, evidencing the rearrangement of surfactants adsorbed on glass surfaces. Our results show that counterions have a significant effect on surfactant behavior at the interface. The greater binding affinity of bromide ions to CTA+ micelle surfaces reduces the repulsion among surfactant head groups and enhances the mobility of micelles adsorbed on the interface. Our study provides valuable insights into the behavior of surfactants at interfaces and highlights the potential of optical tweezers for surfactant research. The development of this trap-and-track approach can have important implications for various applications, including drug delivery and nanomaterials.

## 1. Introduction

Optical tweezers have become a powerful tool for remotely and non-invasively manipulating objects on the micrometer to nanometer scales [1,2,3,4]. They trap small objects using the forces generated by a highly focused laser beam [2], which—in spite of being minuscule and barely disturbing the environment—can control the position and motion of microscopic objects. Almost immediately after their invention [5], optical tweezers demonstrated great success in biological studies [6,7,8] and also became an indispensable tool in colloid and interface sciences [9], as well as many other branches of physics [10,11,12,13,14].

Optical tweezers have an interesting application in molecular chemistry when combined with a technique known as single-particle tracking [15,16,17]. In single-particle tracking, the motion of a particle is first observed, and the particle’s dynamics are analyzed from its trajectory, typically by using statistical methods [18,19,20,21,22]. These dynamics can reveal information about the particle’s interaction with its surroundings [17,23]. With the help of optical tweezers, one can observe the motion of a trapped particle instead of the random diffusion of a free particle [12,13,24]. The greatest advantage of this trap-and-track approach is the ability to confine the particle precisely to the region of interest. The optical force exerted by optical tweezers can be easily modeled as a restoring force and directly inserted into the equation of motion, making the statistical analysis of the particle’s motion reliable.

Recently, we have demonstrated the utilization of optical tweezers for single-particle tracking to investigate surfactant behaviors at water-glass interfaces [25,26] (Figure 1). We used gold nanoparticles as an optical probe and analyzed their trajectories on top of surfactant-covered glass surfaces. To study the surfactants adsorbed at the interface, the gold nanoparticles’ vertical movements were confined to the interfacial area by the radiation pressure of the laser beam propagating from the top to the bottom, as illustrated in Figure 1. Their lateral movements were also confined by the so-called gradient force of the focused laser beam, which pulls the particles toward the intensity maximum. This confinement enabled long-term and real-time observation of particle–surfactant interactions, which revealed an active rearrangement of spherical admicelles upon external stimulus (i.e., the presence of gold nanoparticle) [25] and the fusion of bilayer membranes occurring over tens of seconds [26]. However, since we only examined one type of surfactant (cetyltrimethylammonium chloride, CTAC) in these studies, additional experiments with a different type of surfactant can strengthen the general applicability of optical tweezers for surfactant characterization.

In this communication, we first introduce to the Materials Chemistry Community optical tweezers as a new tool for characterizing surfactants adsorbed at interfaces. This method has the distinct advantage of investigating the local interactions between surfactants and an external stimulus brought by the optical tweezers, which are often inaccessible with classical bulk or ensemble measurements. We then investigate cetyltrimethylammonium bromide (CTAB) as another example surfactant with optical tweezers. CTAB is a quaternary ammonium surfactant and has diverse applications, including DNA extraction [27], nanoparticle synthesis [28,29], and anticancer agent [30]. Comparing the results of CTAB with those of CTAC, the corresponding chloride salt, we report the primary effect of counterions on surfactant behaviors adsorbed at water–glass interfaces. In both cases, we assume that the surfactants form spherical admicelles at the interfaces [31], which can migrate across surfaces when an optically trapped particle approaches [25]. The comparison between CTAB and CTAC sheds new light on the role of counterions in the mobility of admicelles.

## 2. Results and Discussion

### 2.1. Mean Squared Displacement and Its Correlation with a Surfactant Coverage

The amount of surfactant coverage affects how an optically trapped particle moves on a surface [25]. To understand this dependence, we will first discuss the cationic surfactant’s adsorption isotherm at the solid–aqueous interface and its relationship with a statistical measure of the spatial extent of random motion, which is called the mean squared displacement.

The adsorption of cationic surfactants on oxide surfaces has been extensively studied, and two interactions are known to control the adsorption: electrostatic and hydrophobic interactions [32]. Based on the concentration ranges defined by Atkin et al. [32], the adsorption isotherm of cationic surfactants can be divided into four concentration ranges as illustrated in Figure 2a. Each range is described as follows:IElectrostatic concentration range;Surfactant molecules are electrostatically adsorbed to charged surface sites.Near the initially charged sites, the adsorbed cationic head groups generate additional charged sites.IIElectrostatic and hydrophobic concentration range;Adsorption is driven by both hydrophobic interactions among surfactant tails and the electrostatic attraction.The adsorbed morphology is described as a “teepee” structure (Figure 2a, II).At the end of this concentration range, the substrate ionization is at its maximum, and the overall surface charge is neutralized.IIIHydrophobic concentration range (below the critical micelle; concentration, CMC)Hydrophobic interactions are the sole driving force, and surfactant molecules adsorb to the “teepees” with their head groups facing away from the surface (Figure 2a, III).This globular micellar structure is referred to as an admicelle.The level of counterion adsorption increases noticeably.IVHydrophobic concentration range (above the CMC);Micelles in the solution can directly adsorb to the interface.The surface coverage reaches a plateau, indicating that the surface is fully covered with admicelles.

**Figure 2 molecules-28-02859-f002:**
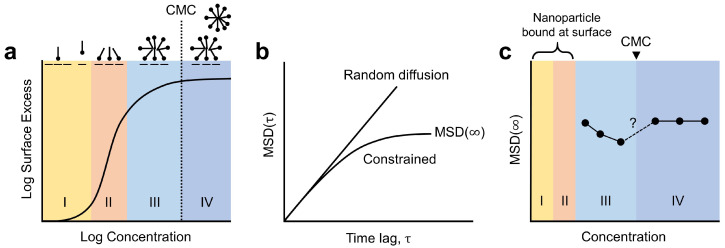
Correlation between adsorption isotherm and mean squared displacement (MSD) of an optical probe at an interface. (**a**) Adsorption isotherm for cationic surfactants on oxide surfaces. Adapted with permission from Ref. [32] Copyright 2003, Elsevier. Each range (I, II, III, and IV) is described in Section 2.1. (**b**) Typical MSDs for a random diffusion and a constrained movement (corresponding to a particle in an optical trap). (**c**) Trends for the MSD plateau values, MSD (*∞*), varying with concentration. Adapted with permission from Ref. [25] Copyright 2021, American Chemical Society.

When gold nanoparticles, which are the optical probes in our trap-and-track experiments, exist in a surfactant solution, surfactants also adsorb on their gold surfaces. They first form a monolayer on gold surfaces and then gradually transform into a bilayer as the concentration increases [33]. For concentration ranges I and II, we found the gold nanoparticles readily stuck on the glass surface following the exact mechanism of surfactant adsorption, i.e., electrostatic attraction and hydrophobic interaction. For concentration ranges III and IV (>0.1 mM for both CTAB and CTAC [25,31]), where the glass surface charge was neutralized, we achieved stable trapping of gold nanoparticles on the surfaces.

In the trap-and-track experiments, the resulting trajectories are statistically analyzed. The most common measure in analyzing random motion is mean squared displacement (MSD) [18,19,20,21,22]. For time-series position data of length *N*, a trajectory $T_{N}$ is expressed as
(1)$$T_{N}=\left\{\left(X_{1},Y_{1}\right),\left(X_{2},Y_{2}\right),\ldots ,\left(X_{N},Y_{N}\right)\right\}\;,$$
where the gold nanoparticle center position is defined by the *X* and *Y* coordinates on the glass surface plane. The time-averaged MSD for this trajectory $T_{N}$ is defined as follows [34]:(2)$$\;\mathrm{MSD}\left(\tau \right)=\frac{1}{N-\tau }\sum_{i=1}^{N-\tau }\{{\left(X_{i+\tau }-X_{i}\right)}^{2}+{\left(Y_{i+\tau }-Y_{i}\right)}^{2}\}\;,$$
for any time lag τ=1,2,…,N−1. This time lag τ refers to a temporal window, and the MSD can be considered as the region of a system explored by the random walker within this temporal window. As illustrated in Figure 2b, for a purely random walk, the MSD increases linearly with τ. For a tracer in an optical trap, the MSD reaches a plateau for longer time lags because the optical tweezer limits how far the particle can travel. We call this plateau value MSD(*∞*) and use it as a representative value for a measured trajectory.

In our recent work, we discovered an interesting relationship between MSD(*∞*) and surfactant coverages [25]. In the concentration range III, where the glass surface was partially covered with admicelles, the MSD(*∞*) decreased with the increasing concentration, indicating that the particles were confined more tightly. Above the solution critical micelle concentration (CMC), on the other hand, the MSD(*∞*) recovered its value from the lowest concentration (Figure 2c). We interpreted this trend with the emergence of a new type of trapping potential: the electrostatic trapping potential produced by rearranging admicelles. In general, admicelles are mobile and can migrate over the surface, a process that requires significantly less energy than desorption [35]. Electrostatic repulsion between the particle and admicelles and thermal energy provided by the optically heated gold nanoparticle can be the primary driving force in the rearrangement mechanism [25]. The resulting electrostatic trapping potential becomes deeper and narrower as the coverage expands, but it vanishes once the surface is completely covered. This interpretation corresponds closely to the experimentally observed MSD(*∞*). A limitation of the study was the relatively long concentration interval of 0.5 mM, which obscured detailed changes, especially near the CMC (Figure 2c).

### 2.2. Comparison between CTAC and CTAB

As an extension of our recent work on CTAC, we aim to address here the effect of counterions (Cl${}^{-}$ and Br${}^{-}$) on the kinetic behavior of admicelles by investigating MSD(*∞*) with CTAB. In particular, we examine the trapped particles’ behaviors near the CMC with a reduced concentration interval, revealing detailed changes in MSD(*∞*).

Figure 3a shows exemplary MSD curves for a 150 nm gold nanoparticle at three different CTAB concentrations (0.1, 0.7, and 1.3 mM). The gold nanoparticles were optically trapped close to the glass–aqueous interface to promote particle–admicelle interactions. As illustrated in Figure 2b, the MSD curves exhibit a typical plateau at their long-time limits (MSD(*∞*)), corresponding to the area where the particles’ motions are confined. Here, we reaffirm a strong dependence of MSD on surfactant concentration.

Figure 3b summarizes MSD(*∞*) as a function of CTAB concentration. We recorded at least five particle trajectories at different locations for each concentration near the CMC of CTAB (0.9 mM). From 0.7 mM to 1.3 mM, we measured the trajectories with a 0.1 mM concentration interval. Compared to the results with CTAC in our earlier study (Figure 3c), the most noticeable feature is that they exhibit the same decreasing trend below the CMC and then recover the previous level above the CMC. This consistent pattern with CTAB and CTAC supports our interpretation of the rearrangement of admicelles upon particle intervention. Furthermore, since we used a smaller concentration interval close to the CMC, we captured the recovery of MSD(*∞*) right below the CMC (Figure 3b). According to this recovery, above a certain coverage, the electrostatic trapping potential becomes too narrow (possibly smaller than the particle footprint), which reduces its contribution to the particle movement and thus causes a gradual transition observed from 0.7 mM to 0.9 mM.

In addition to this similarity, we also found a general downward shift in the MSD(*∞*) when comparing Figure 3b,c. This is related to the effect of the counterions (Br${}^{-}$ and Cl${}^{-}$) since the surfactant molecules (CTA${}^{+}$) in CTAB and CTAC are identical. It is well known that counterions play an essential role in both the self-assembly and adsorption mechanisms of surfactant molecules [36,37,38]. In bulk solutions, counterions bind to surfactant micelle surfaces and screen the electrostatic repulsion between the ionic headgroups, stabilizing the micelles. The binding affinity of Br${}^{-}$ to CTA${}^{+}$ is known to be five times greater than that of Cl${}^{-}$ [39], which results in a lower CMC of CTAB (0.9 mM) than that of CTAC (1.4 mM). The counterions also influence the adsorption of ionic surfactants; Velegol et al. [38] reported a 60% increase in the surface excess of CTA${}^{+}$ on silica when they changed the counterions from chloride to bromide ions. Based on this understanding, we can speculate that the greater binding affinity of Br${}^{-}$ to CTA${}^{+}$ can more efficiently mediate the interactions between a trapped particle and admicelles and make it easier for admicelles to move across surfaces. In particular, the more prominent decrease in MSD(*∞*) below the CMC in Figure 3b indicates that the Br${}^{-}$ facilitates the rearrangements of admicelles across glass surfaces by penetrating more deeply into the surface of the admicelles [37].

## 3. Materials and Methods

### 3.1. Materials

Cetyltrimethylammonium bromide (CTAB) was obtained from Sigma-Aldrich (St. Louis, MO, USA) (≥99 %, Product Number H6269) and was dissolved in distilled water to form a 0.1 M stock solution. The corresponding chloride salt (CTAC) was also obtained from Sigma-Aldrich in solution (25 wt.% in H${}_{2}$O). The stock solutions were diluted to achieve concentrations below and above their corresponding CMCs. Gold colloids with 150 nm diameter, stabilized in citrate buffer, were purchased from Sigma-Aldrich (a product of CytoDiagnostics, Inc., Burlington, ON, Canada) and used as an optical probe in trapping experiments.

### 3.2. Sample Preparation

A volume of 1 mL stock gold colloids was centrifuged at 200× *g* for 30 min (Fisherbrand GT2R Centrifuge) for separation. After carefully removing the supernatant, the residue was re-dispersed in a surfactant solution using a vortex mixer. At this stage, the particle concentration was adjusted by diluting the particle–surfactant mixture with a particle-free surfactant solution of the same concentration to have a sparse appearance under an optical microscope and thus to ensure single-particle trapping. The approximate particle concentration was about $3.6\times 10^{4}$ particles/µL.

A fluid chamber was constructed using a pair of borosilicate glass coverslips (145 µm in thickness) and a double-sided adhesive spacer (120 µm in thickness, Grace Bio-Labs SecureSeal^™^ imaging spacer). We slightly overfilled the chamber with the dilute particle-surfactant mixture to minimize the air bubbles trapped inside the chamber. To prevent the liquids from evaporating, all trapping experiments were conducted within the fluid chamber. All glass coverslips were sonicated in acetone and isopropyl alcohol baths for 30 min each before use.

### 3.3. Optical Trapping and Trajectory Recording

A focused beam for optical trapping was generated using a He-Ne laser and a dry objective lens (60×, 0.85 NA). The laser beam was passed through a particle-containing fluidic chamber installed on a commercial optical microscope (IX71, Olympus), with which optically trapped particles were imaged. Details on the trapping and imaging setup can be found in our previous publications [25,26]. Time-lapse image data were recorded with a CMOS camera (CM3-U3-50S5C-CS, FLIR) at a frame rate of 300 frames per second. The trajectory of a particle was extracted from the recorded videos using a Python package, Trackpy [40], which is based on feature-finding and linking algorithms developed by Crocker and Grier [41].

## 4. Conclusions

In conclusion, this study demonstrates the effectiveness of optical tweezers combined with trajectory analysis as a valuable tool for investigating the local interactions between surfactants and an external stimulus at interfaces. Through the trap-and-track study of CTAB at water–glass interfaces and its comparison with CTAC, we were able to reveal the effect of counterions on surfactant behavior. The greater binding affinity of bromide ions to CTA${}^{+}$ micelle surfaces reduces the repulsion among surfactant head groups, leading to the enhanced mobility of admicelles. These findings not only contribute to the understanding of counterion effects on surfactant behaviors, but also extend the general applicability of optical tweezers in various surfactant research.

Furthermore, the proposed method can be used to study surfactants with different head groups or tail structures, such as gemini surfactants, to advance our understanding of interfacial mobility with different surfactant structures. Additionally, the trap-and-track method can be used to elucidate the effect of electrolytes or complex anions on surfactant behaviors at interfaces.

Overall, this study highlights the potential of optical tweezers as a powerful tool for characterizing surfactants and providing pinpoint access to the microscopic world. By shedding light on the complex behavior of surfactants at interfaces, this research paves the way for future advancements in the field of surfactant research. 

## Figures and Tables

**Figure 1 molecules-28-02859-f001:**
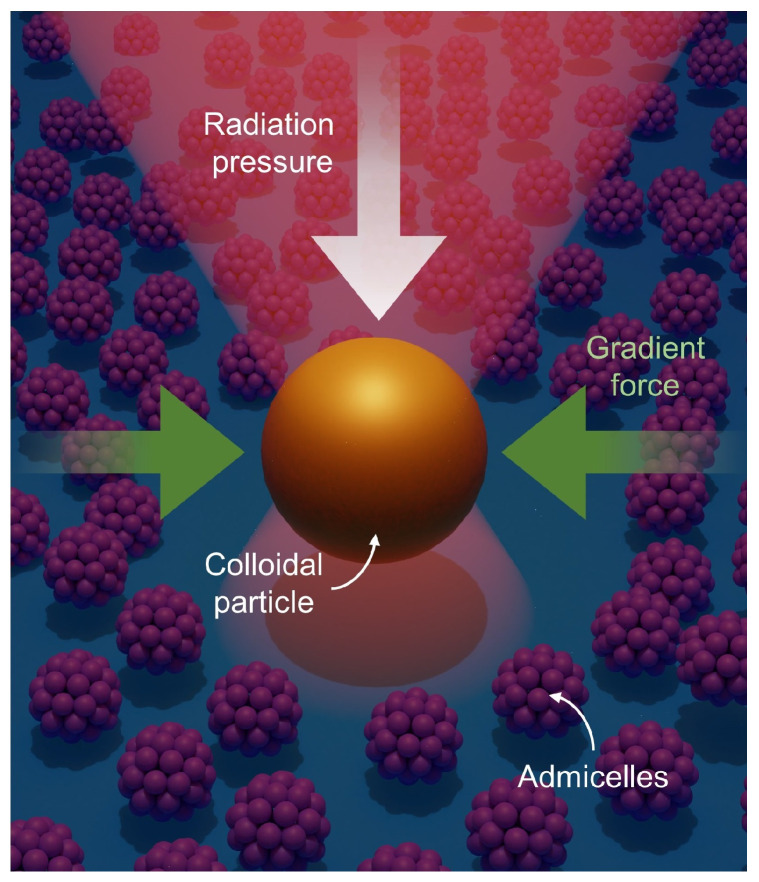
Illustration of a gold nanoparticle optically trapped at a micelle-covered interface. Within a tightly focused laser beam, the particle experiences two distinct forces: radiation pressure and a gradient force. The radiation pressure pushes the particle along the laser beam’s propagation direction (from the top to the bottom surface in the illustration). In contrast, the gradient force produced by a large intensity gradient near the laser focus attracts the particle toward the intensity maximum.

**Figure 3 molecules-28-02859-f003:**
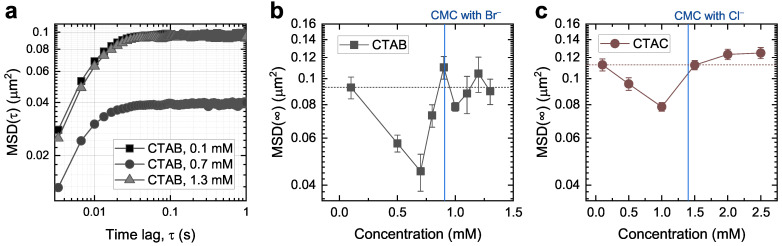
MSD for 150 nm gold nanoparticles optically trapped at the glass–aqueous interface in surfactant solutions. (**a**) Double-logarithmic plot of the MSD at three different CTAB concentrations. (**b**,**c**) Long-time limits, MSD(*∞*), as a function of CTAB (**b**) and CTAC (**c**) concentration near the CMC. The symbols represent the mean values of at least five measurements, and the error bars are the standard error of the mean. The corresponding CMCs are indicated as vertical blue lines, and the MSD(*∞*) at the lowest concentration is noted as horizontal dotted lines as a reference.

## Data Availability

The data presented in this study are openly available in Zenodo at DOI:10.5281/zenodo.7685023, reference number 7685024.

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
