# Peer review of "Trap-and-Track for Characterizing Surfactants at Interfaces"

_molecules, 2023, doi:10.3390/molecules28062859_

Round 1

Reviewer 1 Report

Thank you for your interesting work. Would be interesting to have data closer to region II and to evaluate effect of ionic strength.

See comments

Author Response

We thank the Reviewer for their careful reading and positive assessment of our work. By answering the following comments, we address the measurements closer to region II and the effect of ionic strength.

Response to Reviewer 1 comments

Line 6 in Abstract: ‘and study the surfactant’s response to this external stimulus by statistically analyzing the particle’s movement.’

Not response of surfactant, but that of trapped tracer particles on change of surfactant concentration is measured.

Response: It is true that we measured the movement of trapped tracer particles in response to surfactant concentration. However, by analyzing the movements, we eventually evidenced the rearrangement of adsorbed surfactants, which can be stated as a response of surfactants to the external stimulus (i.e., the presence of a trapped gold nanoparticle). The Reviewer’s comment caught our attention that the corresponding expression can be misleading, and therefore, we revised it as follows:

We use optical tweezers to trap a gold nanoparticle and statistically analyze the particle’s movement in response to various surfactant concentrations, evidencing the rearrangement of surfactants adsorbed on glass surfaces.

Line 73 in Section 2.2: ‘the particle concentration was adjusted by diluting the particle-surfactant mixture with a particle-free surfactant solution of the same concentration to have a sparse appearance under an optical microscope and thus to ensure single-particle trapping.’

Reason is understandable, however, it would be good to give an approximate particle concentration.

Response: We are grateful to the Reviewer for pointing out this missing information. We added the particle concentration in the revised manuscript:

… to ensure single-particle trapping. The approximate particle concentration was about 3.6 ×104 particles/μL.

Line 127 in Section 3: ‘They first form a monolayer on gold surfaces and then gradually transform into a bilayer as the concentration increases [35].’

This is indeed likely the case, given a dense cover by citrate, different from glass and mineral oxide surfaces. However, in [35] it is not proved by their results, which could be explained in their case also by formation of admicelles too.

Response: In ref. [35], Li et al. provided a possible evolution of CTAB assemblies on citrate-capped gold nanoparticle surfaces. Based on their experimental evidence with UV-vis spectra, zeta potentials, and TEM images, they suggested that the CTAB assemblies undergo transitions in their structures with increasing surfactant concentration as follows: (incomplete monolayer) → (complete monolayer) → (incomplete bilayer) → (complete bilayer) → (bilayer + strongly associated micelles), which is summarized in their Fig. 3 of ref. [35]. In addition, the assembly structures were strongly dependent on the particle-surfactant ratio, not only the surfactant concentration.

We measured the hydrodynamic sizes and zeta potentials of gold nanoparticles in our previous study in ref. [25] and  identified the concentration ranges corresponding to each assembly stage for gold nanoparticles used in this study. Below the CMC, the particle experiences transitions from imperfect bilayer to perfect bilayer as mentioned in the manuscript. Above the CMC, the bulk micelles start to be strongly associated on top of the bilayer surfaces.

Fig. 2: ‘no trapping’ should be exchanged by nanoparticle bound at surface. (However, a sufficiently low concentration surface and particles both should be negatively charged!) Suggest to better highlight the difference between particle adsorption on glass and trapping close to the interface.

Response: We appreciate the Reviewer's input on the idea in Fig. 2. The expression “no trapping” was changed to "nanoparticle bound at surface" in the revised manuscript. Indeed, at a sufficiently low surfactant concentration, e.g., without the addition of surfactants, both the particle and glass surfaces are negatively charged, and we were able to perform trapping experiments. However, the problem in that case was that the colloidal nanoparticle concentration could not be lowered enough to ensure single-particle trapping due to the aggregation of particles when diluted with water.

(Continued) Seems to me adsorbed particles have almost zero displacement. Why does MSD not start from 0 in region III?

Response: According to the definition in Eq. 2, MSD for adsorbed particles is indeed equal to zero. The MSD, however, is only meaningful when the particle is moving. We can infer factors influencing the particle's motion, such as a trapping force or fluid viscosity, from MSD as a function of time lag. We cannot formulate the equation of motion and cannot perform any additional analysis for a stuck particle. As a result, we limited our MSD analysis to "trapped" particles exhibiting thermal motions. 

(Continued) Due to different surface properties of glass and NP, at given surfactant concentration surface coverage should be different. What effects do you expect from this?

Response: Surface coverage and assembly structure greatly influence the interactions between a trapped particle and the nearby surface. In our previous study in ref. [25], we extensively discussed how this different surfactant coverage might affect the interaction between a trapped tracer and the glass wall. Our findings suggest that moderate interactions, such as admicelle rearrangements, occur in heterogeneous assemblies (globular admicelles on glass surfaces and bilayer on gold surfaces), making coverage differences less noticeable. In contrast, homogeneous assemblies in another study (ref. [26]) exhibit more radical interactions, such as membrane fusion. Notably, in this study, the maximum surface coverages on the glass and gold nanoparticles surfaces occur around the CMC, making the surface coverage on the two surfaces coincidentally similar. We anticipate that the impact of different coverages on heterogeneous surfaces can change surface interactions, so it is important to assess this possibility.

(Continued) As ionic strength varies with surfactant concentration, how does this influence observed variations?

Response: Changes in ionic strength may impact the distance between a trapped nanoparticle and a nearby surface by altering the electrostatic interaction. In our previous study (ref. [25]), we found experimental evidence indicating that the drag coefficient, which depends on the particle-glass distance, remains consistent across the whole concentration range. This observation was surprising, as we had anticipated that the particle's vertical position would be determined by the balance between radiation pressure from the laser beam and electrostatic repulsion between the charged surfaces, which changes with surfactant concentration. One possible explanation for this result is that the surfactant molecules adsorbed at the contact area may migrate outside of this region due to electrostatic repulsion, which could account for the fact that overall ionic strength does not determine the particle height.

Line 197 in Section 3.2: ‘the interactions between a trapped particle and admicelles and make it easier for admicelles to move across surfaces.’

Response: We thank the Reviewer bringing this error to our attention. We made the suggested revisions.

Reviewer 2 Report

In this manuscript, innovative possibilities in cationic surfactants characterization are presented and elaborated. The effect of counterions on surfactant behaviors adsorbed at water-glass interfaces is elaborated in detail.

The manuscript is correctly written, scientifically sound, and clearly presented. Employed experimental methods are adequate and correctly interpreted to support the conclusions. Relevant issues are adequately discussed. 

Since cationic surfactants differ much in their structure, different surfactants with some other head groups, different numbers of tails, and possible complex anions can be studied with an elaborated method also. Can authors give some explanation of possible results in this situation? 

I don’t have any other comments, presented work is excellent and has wide possibilities for application in different branches of chemistry related to surfactants, such as the characterization of new surfactants, a study of the interaction of surfactants and capillaries in electrophoresis, etc. 

Author Response

We thank the Reviewer for their positive assessment of our manuscript and their constructive comments.

Response to Reviewer 2 comments

Point 1:  Since cationic surfactants differ much in their structure, different surfactants with some other head groups, different numbers of tails, and possible complex anions can be studied with an elaborated method also. Can authors give some explanation of possible results in this situation?

Response: Studying a range of different surfactants with the variations the Reviewer mentioned can provide valuable information about how the structure of the surfactant affects its properties and behavior at interfaces.

  • Other head groups: The study of sodium dodecyl sulfate (SDS) with the presented trap-and-track method would be interesting. SDS is a common cationic surfactant used for DNA or RNA extraction and is widely applied in the pharmaceutical industry. The CMC of SDS in water at 25°C is 8.2 mM, which is significantly higher than that of CTAB or CTAC. Comparing SDS with CTAB, for instance, would provide a better understanding of interfacial mobility with different head groups.
  • Different numbers of tails: Gemini surfactants are yet another intriguing potential research subject. They are composed of two conventional surfactant molecules that are chemically joined together by a spacer, and one can precisely control the mobility and packing geometry of the micelle by varying the length of the spacer. Future research using the trap-and-track method may help to yield important information about the properties and behavior of these molecules.
  • Complex anions: Instead of changing surfactant concentrations, one can also add electrolytes to a fixed surfactant concentration solution and study the particle behaviors. The addition of electrolytes would decrease repulsions between adsorbed surfactant head groups, and one can elucidate the effect of electrolytes on admicelle behaviors at interfaces.

Thanks to the Reviewer’s comment, we have added a short discussion on these potential applications in the Conclusions of the revised manuscript.

Point 2: I don’t have any other comments, presented work is excellent and has wide possibilities for application in different branches of chemistry related to surfactants, such as the characterization of new surfactants, a study of the interaction of surfactants and capillaries in electrophoresis, etc.

Response: We are grateful to the Reviewer’s positive feedback on our work. The Reviewer’s mention of the characterization of new surfactants and the study of surfactant/capillary interactions in electrophoresis are both excellent points, and we agree that these areas could benefit from the methods presented in our paper.